# Bayesian uncertainty quantification of engineering models for wind-farm atmosphere interaction

Frederik Aerts<sup>1</sup>, Koen Devesse<sup>1</sup>, and Johan Meyers<sup>1</sup>

<sup>1</sup>Department of Mechanical Engineering, KU Leuven, Celestijnenlaan 300, B-3001 Leuven, Belgium

**Correspondence:** Frederik Aerts (frederik.aerts1@kuleuven.be)

**Abstract.** Accurate modeling of wind-farm atmosphere interactions is critical for reliable energy yield assessments and flow control strategies. However, formal model comparison methodologies that quantify model form uncertainty by also accounting for parameter uncertainty are still lacking. This study presents an enhanced Bayesian uncertainty quantification framework for the calibration and validation of engineering wind farm flow models. Building on previous work, the framework explicitly incorporates model inadequacy through a parameterized model error distribution, enabling the separation of model and measurement uncertainties. The improved framework is demonstrated using a large-eddy simulation dataset for wind-farm blockage and atmospheric gravity waves in conventionally neutral boundary layers. Two models of differing fidelity – a standard Gaussian wake model and an atmospheric perturbation model (APM) – are calibrated and compared. The posterior distribution of the model parameters reveals insights into model behavior and highlights areas for further improvement, for instance, when estimated parameter values are inconsistent across the model chain. In addition, it is shown that not explicitly incorporating model inadequacy results in an overly confident posterior distribution, and renders derived stochastic flow models incapable of representing model uncertainty. A comparison of the quantified model uncertainty shows that the APM has significantly lower uncertainty than a standard wake model for this dataset, as the wake model is unable to represent wind-farm blockage effects. This demonstrates the utility of the framework for objective model comparison with quantified parameter and model uncertainty given a reference dataset. Both the framework and the parallelized Sequential Monte Carlo algorithm for accelerated posterior sampling are made available through the open-source Python package UMBRA.

#### 1 Introduction

Wind farm flow model bias and uncertainty directly impact both the profitability of wind projects through pre-construction energy yield assessments (Lee and Fields, 2021) and the financial targets of wind developers through production forecasts (Ørsted, 2019). Although the estimation bias in annual energy predictions (AEP) has steadily declined in the last two decades, the uncertainty remained of similar magnitude (Lee and Fields, 2021). Historically, wind farm performance is one of the largest contributors to AEP uncertainty, in part due to the uncertainty of the power losses on downstream turbines due to turbine wake effects (Clifton et al., 2016; Lee and Fields, 2021). Although models for these wake losses can reproduce trends in benchmark observations, their precision is highly variable; motivating the use of more precise and delineated observations of wind farms under many different operating and atmospheric conditions instead of those averaged over long periods of time (Moriarty et al.,

2014). Moreover, the increasing capacity density and size of wind turbines require new wind farm flow models that consider the atmospheric boundary layer from the surface to the free atmosphere to model effects such as wind-farm blockage (Allaerts and Meyers, 2019) and wakes (Bastankhah et al., 2024). To turn new wind farm flow models into reliable and cost-effective tools, objective methods are needed to validate them, quantify their uncertainty, and eventually calibrate them with a wide variety of flow conditions (Rodrigo et al., 2017). The present manuscript studies the use of Bayesian uncertainty quantification as an objective method for model comparison given a reference dataset and calibration with quantified parameter uncertainty.

Many small- to large-scale benchmarking studies have validated the suitability of wind turbine wake models to represent energy losses in downstream turbines with historical power data (see Doekemeijer et al. (2022) for a comprehensive overview). Typically, these studies compare metrics such as farm power, turbine power for a given bin of wind directions, and wake loss, while using default wake model parameters (Doekemeijer et al., 2022). This is common practice, as it is the most objective way of quantifying the baseline performance of the models (Rodrigo et al., 2017). However, it is known that the wake model bias is site-specific (Nygaard, 2015) and that the wake recovery differs between offshore and onshore wind farms (Barthelmie et al., 2009; Göçmen et al., 2016) and with atmospheric conditions (Abkar and Porté-Agel, 2015; Niayifar and Porté-Agel, 2016; Klemmer and Howland, 2025). Therefore, site-specific model tuning is crucial for accurate production forecasts. Moreover, in wind farm flow control, the flow model may be tuned to site-specific data in an open- or closed-loop fashion to adequately represent the flow field at any time (Göçmen et al., 2022; Meyers et al., 2022). Therefore, any model validation procedure may benefit from including model calibration so that model performance in practical applications can be compared objectively.

Concerning the quantification of uncertainty, one must distinguish between forward (data-free) and inverse (data-driven) UQ methods (Xiao and Cinnella, 2019). Forward UQ examines the effect of prespecified uncertainties on model inputs on the model outcome and is widely used to quantify the effect of wind resource variability on AEP estimates (Lackner et al., 2007; Kwon, 2010; Clerc et al., 2012). The procedure of Gaumond et al. (2014) to assess the effect of wind direction uncertainty on the predicted power of wake models also adheres to this approach. Despite being rigorous, forward UQ relies on the estimates of the constituent uncertainties, which may be subjective (Nygaard, 2015). Inverse UQ estimates the uncertainty of the model, and possibly its parameters, by comparing it with measured data. The estimated distribution of the discrepancy between model predictions and measurements determines the model uncertainty through its width and the model bias through its mean. With an inverse UQ using operational data from 19 offshore wind farms, Nygaard et al. (2022) showed that the uncertainty on the predicted wake loss relative to the observed wake loss is less than 10% of the observed loss for the TurboPark model. This is significantly lower than previously estimated (Walker et al., 2015), in part due to thorough data processing and the inclusion of heterogeneous background flow. However, the uncertainty of the model is still overestimated because it is not separated from the experimental uncertainty. With Bayesian inverse UQ, it is possible to separate the measurement and model uncertainty, as demonstrated with operational power data from the Westermost Rough wind farm, while also accounting for the uncertainty of the model parameters (Aerts et al., 2023).

Compared to deterministic model calibration methods (see van Binsbergen et al. (2024) for a comprehensive overview), Bayesian calibration gives not only the 'best' parameters, but a joint posterior distribution with information on the parameter uncertainties, and their correlations given the dataset (Aerts et al., 2023). The posterior parameter distribution may inform

https://doi.org/10.5194/wes-2025-196 Preprint. Discussion started: 22 October 2025

© Author(s) 2025. CC BY 4.0 License.

modelers about missing physics, when parameter values are estimated differently throughout the model chain. For example, LoCascio et al. (2023) showed that the posterior mean wake expansion rate and its uncertainty differ for different wake merging methods, which is a cautionary finding for a modular approach to wake modeling. In addition, Zhang and Zhao (2020) and LoCascio et al. (2023) have proposed using the posterior distribution of the model parameters to obtain stochastic wake models, which can be used in wake steering under uncertainty (Howland, 2021). However, the current approaches to obtain such stochastic models include only the epistemic uncertainty (i.e. due to limited data) on the model parameters, and not the model uncertainty due to varying physical phenomena not captured by the (deterministic) model.

In this study, we improve on a previously developed Bayesian UQ framework (Aerts et al., 2023) and demonstrate its use in a controlled environment with large model uncertainty. To this end, we select a large-eddy simulation dataset for blockage due to atmospheric gravity waves (Lanzilao and Meyers, 2024) as reference data, and perform an inverse UQ for a standard wake model and a recently developed atmospheric perturbation model (Devesse et al., 2024b). Specifically, we present how the model uncertainty can be incorporated into the Bayesian framework to obtain stochastic models that include model uncertainty. In addition, we illustrate how the adequate inclusion of model uncertainty is crucial to obtain a correct posterior distribution of the empirical parameters, which is used in the current stochastic wake models. Lastly, we show how the Bayesian UQ framework can be used for objective model comparison with quantified model and parameter uncertainty. In contrast to previous studies (Zhang and Zhao, 2020; Aerts et al., 2023; LoCascio et al., 2023), which relied on inherently serial Markov chain Monte Carlo algorithms such as (Adaptive) Metropolis Hastings (Haario et al., 2001) and Hamiltonian Monte Carlo with No-U-Turn tuning (Hoffman et al., 2014) to approximate the posterior distribution, we employ an inherently parallel sampling algorithm that still performs so-called 'exact' posterior inference. The parallelized sampler and Bayesian framework are made available in an open-source Python package coined UMBRA: Uncertainty Modeling toolbox for Bayesian data Re-Analysis.

The improved Bayesian UQ framework is presented in Sect. 2, together with the inherently parallel algorithm to sample the posterior. The setup of the demonstration case is introduced in Sect. 3, by presenting the essential parts of the wind farm flow models and dataset. The results of the inverse UQ analyses are presented in Sect. 4, with emphasis on the consequences of neglecting model error (Sect. 4.1), the framework's adequacy (Sect. 4.2), its application to objective model comparison with UQ (Sect. 4.3), and the generalizability of the findings (Sect. 4.4). A summary and outlook are given in Sect. 5.

## 2 Bayesian uncertainty quantification

We first discuss how different sources of uncertainty are included in the UQ framework in Sect. 2.1, building on earlier work (Aerts et al., 2023). Section 2.2 demonstrates how this formulation naturally leads to Bayesian updating from a prior to a posterior distribution, and introduces the posterior predictive distribution as a key tool for validation in Bayesian UQ. The parallelizable algorithm employed for posterior sampling is detailed in Sect. 2.3.

#### 2.1 Sources of uncertainty

Wind farm flow models aim to predict quantities of interest for a given atmospheric state. When focusing on turbine-level power output, the idealized 'true process'  $T(\psi)$  would yield the power  $\mathcal{P} \in \mathbb{R}^{N_T}$  of the  $N_T$  turbines, normalized by the output of an undisturbed upstream turbine, given a complete description of the wind farm and atmospheric state  $\psi$ :

95 
$$\mathcal{P} = T(\psi)$$
, (1)

In practice, this process is approximated by a model  $\mathcal{M}(\vartheta_e,\varphi)$ , which relies on empirical parameters  $\vartheta_e$  and a partial state description  $\varphi$ . This approximation introduces model bias and uncertainty (Sect. 2.1.1). Since the true process is only accessible through measurements, measurement uncertainty also arises (Sect. 2.1.2). Moreover, the most representative values of  $\vartheta_e$  for the given conditions are initially unknown. Section 2.1.3 describes how this uncertainty is represented in a prior probability density.

## 2.1.1 Model error

100

Since the wind farm flow model is typically imperfect, we can define an additive model error  $E_B(\vartheta_e, \psi)$  that depends on the choice of the model parameters  $\vartheta_e$  and the state of the wind farm and atmosphere  $\psi$ :

$$T(\psi) \triangleq \mathcal{M}(\vartheta_e, \varphi) + E_B(\vartheta_e, \psi). \tag{2}$$

Note that the model error is, in fact, deterministic given  $\vartheta_e$  and  $\psi$ , so that  $p(\mathcal{E}_B|\vartheta_e,\psi) = \delta(\mathcal{E}_B - E_B(\vartheta_e,\psi))$ . However, since usually only a subset of the variables describing the atmospheric state is available or included as input to the model, we are interested in the distribution of the true process  $\mathcal{T}$  conditioned on  $\varphi = \psi \setminus \psi'$ , given by

$$p(\mathcal{T}|\varphi) = \int p(\mathcal{T}|\psi)p(\psi'|\varphi) \,d\psi'$$

$$= \int \delta(\mathcal{M}(\vartheta_e, \varphi) + \mathcal{E}_B(\vartheta_e, \psi) - \mathcal{T})p(\psi') \,d\psi'.$$
(3)

The unmodeled or unobserved variations in atmospheric conditions related to  $\psi' = \psi \setminus \varphi$  thus introduce uncertainty in the conditional process  $\mathcal{T}|\varphi$  through the conditional model error  $\mathcal{E}_B|\varphi$ . Note that if we were to take  $\mathcal{T}|\varphi \triangleq \mathcal{M}|\vartheta_e, \varphi + \mathcal{E}_B|\vartheta_e, \varphi$  as a starting point, the aforementioned uncertainty must be reflected in the model itself or the model error. Hence, we have to choose whether to incorporate this uncertainty – which we will further refer to as model uncertainty – within the model parameters, the model error, or both.

The first option is to incorporate the model uncertainty within the model parameters (see also Sargsyan et al., 2015; Wu et al., 2018a). To that end, we consider that each realization  $(\psi_i, \mathcal{T}_i)$  is associated with model parameter values  $\vartheta_{e,i}$  that yield the same model error, i.e.,  $E_B(\vartheta_{e,i}, \psi_i) = \mu_B$ . Since only  $\varphi_i$  is known, the resulting uncertainty on the model output is quantified by propagating the distribution of the parameters  $\vartheta_{e,i}$  through the model. For a normal distribution  $\vartheta_{e,i} \sim \mathcal{N}(\mu_{\vartheta}, \sigma_{\vartheta}^2)$ , the

resulting distribution of the model output in terms of  $\mu_{\vartheta}$  and  $\sigma_{\vartheta}$  is

$$p(\mathcal{T}_{i}|\mu_{\vartheta}, \sigma_{\vartheta}, \varphi_{i}) = \int p(\mathcal{T}|\vartheta_{e,i}, \varphi_{i}) p(\vartheta_{e,i}|\mu_{\vartheta}, \sigma_{\vartheta}) \, d\vartheta_{e,i}$$

$$= \int \delta(\mathcal{M}(\vartheta_{e,i}, \varphi_{i}) + \mu_{B} - \mathcal{T}_{i}) \mathcal{N}(\vartheta_{e,i}; \mu_{\vartheta}, \sigma_{\vartheta}^{2}) \, d\vartheta_{e,i}$$

$$\approx \mathcal{N}(\mathcal{T}_{i}; \mathcal{M}(\mu_{\vartheta}, \varphi_{i}) + \mu_{B}, \sigma_{\vartheta}^{2} J_{\mathcal{M}} J_{\mathcal{M}}^{\top})$$

$$(4)$$

where the approximation corresponds to a linearization of the model  $\mathcal{M}(\vartheta_e,\varphi) \approx \mathcal{M}(\mu_\vartheta,\varphi) + J_\mathcal{M}(\vartheta_e - \mu_\vartheta)$  for a scalar parameter  $\vartheta_e \in \mathbb{R}$  and Jacobian  $J_\mathcal{M} \in \mathbb{R}^{N_T}$ . Hence the uncertainty is determined by the model parameter uncertainty  $\sigma_\vartheta$  and the model structure through the Jacobian  $J_\mathcal{M}$ . However, the model uncertainty may have a different structure than the model itself. Taking the example of a wake model with an unknown wake expansion rate, the model uncertainty on the upstream turbines due to blockage cannot be captured in this approach as the sensitivity of the upstream turbine power to the wake expansion rate is zero. Hence an additional model error term is generally needed. Moreover, a forward UQ is still necessary to translate the uncertainty on the model parameters to uncertainty on the model output, which is typically of interest for many practical applications.

The second approach attributes the model uncertainty to the model error term. To that end, we select each observation  $(\psi_i, \mathcal{T}_i)$  to have a corresponding and distinct model error  $E_B(\vartheta_{e,i}, \psi_i)$ , which is not necessarily the same for all observations, given the fixed model parameter values  $\vartheta_e$  that best represent all observations. Hence, the model uncertainty is represented in the distribution of the model error  $p(\mathcal{E}_B|\varphi)$ , obtained by the marginalization

$$p(\mathcal{E}_B|\varphi) = \int p(\mathcal{E}_B|\psi)p(\psi') \,d\psi' \approx \mathcal{N}(\mathcal{E}_B; \mu_B, \Sigma_B)$$
 (5)

where we presume a normal distribution of the model error, which is the maximum entropy distribution given that we are only interested in the first and second order moments (McElreath, 2018). This approach is independent of the model structure and is readily interpreted as uncertainty on either the model or the model error due to additivity. Note that if both approaches are combined, there is no way to distinguish uncertainty that would result from the model parameter(s) or from the additive term. To allow for a general model error uncertainty parameterization while avoiding that the problem becomes underdetermined, we opt for the second approach. Hence, the model error distribution is parameterized by its expected value  $\mu_B$  and covariance  $\Sigma_B$ .

The current parameterization of the model error scales with the number of turbines  $N_T$  in the farm since  $\mu_B \in \mathbb{R}^{N_T}$  and  $\Sigma_B \in \mathbb{R}^{N_T \times N_T}$ . To reduce its dimensionality, we follow the same approach as in our previous work (Aerts et al., 2023). The correlations in  $\Sigma_B$  are neglected and the mean and standard deviation of the model error on the power of the turbine i is binned based on the number of upstream turbines  $\zeta(i)$  that cause a wake loss on the turbine i greater than 1% of the free stream wind speed. Hence,

$$145 \mu_{B,i} = \delta_{\zeta(i)} (6)$$

$$\Sigma_{B,ij} = \delta_{ij} \sigma_{B,\zeta(i)}^2 \tag{7}$$

150

with  $\delta_{ij}$  the Kronecker delta. As such, the model error distribution is parameterized by the model error parameters  $\vartheta_b = \{\delta_\zeta, \sigma_{B,\zeta}\}_{\zeta=0}^{\zeta_{max}}$  with  $\zeta=0$  the index for the upstream turbines,  $\zeta=1$  the index for turbines with one upstream turbine that wakes them, and so forth.

## 2.1.2 Measurement error

The true process can only be observed with measurements  $\mathcal{P}^*$ , which come with a measurement error  $\mathcal{E}_M$ 

$$\mathcal{P}^* = \mathcal{T}|\psi + \mathcal{E}_M|\psi. \tag{8}$$

We will make the simplifying assumption that the measurement error is independent of the unobserved or unmodelled physics  $\psi'$  such that  $\mathcal{E}_M|\psi\approx\mathcal{E}_M|\varphi$  (Aerts et al., 2023). Since most engineering wind farm flow models represent stationary atmospheric flows, their predictions should be compared with time-averaged data. However, observational and high-fidelity simulation data are typically subjected to temporally resolved turbulence. Because only a finite time period is available for averaging due to changing atmospheric conditions or computational constraints, the measurement error consists both of the error of the apparatus and the averaging error. The measurement bias and standard deviation due to the apparatus are typically known a priori, such that the distribution of the apparatus error can be taken as a normal distribution based on these quantities. In what follows, we will presume that the averaging error dominates. This is certainly true for simulation data, as is the case in our manuscript. Due to the central limit theorem and given that the estimator is unbiased, the measurement error then follows a multivariate normal distribution with zero mean and a covariance matrix  $\Sigma_T$ .

The averaging error covariance matrix  $\Sigma_T$  on the mean can either be prespecified or unknown. For N independent measurements of the power output of each turbine i for the same atmospheric condition, the error on the average of the N measurements can be estimated as  $\mathbb{V}(\mathcal{P}_i^*)/N$  with  $\mathbb{V}(\mathcal{P}_i^*)$  the sample variance (Wasserman, 2013). Equivalently, all individual measurements can be used with the sample variance, given that they are independent and represent the same atmospheric state. For a correlated time series, the moving block bootstrap can be employed (Garcia et al., 2005). If no information is available, the averaging error covariance can also be estimated directly from the data based on a parameterization with parameters  $\vartheta_t$  (Aerts et al., 2023). However, if the covariance structures between the averaging error and the model error are not sufficiently different, they are indistinguishable. Therefore, it is preferred to use the estimated error on the mean if available.

In practice, the inflow conditions can also be uncertain due to any kind of measurement error. This inflow uncertainty may be propagated through the model with a marginalization similar to that in Eq. (4). Note that this procedure is similar to that of Gaumond et al. (2014) to incorporate the effect of wind direction variability on the mean power, but also includes the resulting variance of the model output. In that manner, a part of the total variance in the data can be attributed to inflow uncertainty, thereby reducing the observed model uncertainty.

## 2.1.3 Prior parameter uncertainty

The most representative empirical model parameters of the wind farm flow model  $\vartheta_e$  are a priori uncertain, but the associated model error distribution, parameterized by  $\vartheta_b$ , and possibly measurement error covariance, parameterized by  $\vartheta_t$ , are as well.

This a priori uncertainty can be quantified or specified in a prior distribution  $p(\vartheta_e, \vartheta_b, \vartheta_t)$ , which reflects one's assumptions and state of knowledge before data come along (Trotta, 2008). We will use a weakly informative prior, which is designed to regularize inferences with structural information (Gelman et al., 2017). The provided information is intentionally weaker than any actual prior knowledge available (Gelman et al., 2013) and we choose the shape of the distribution to have the highest entropy given the provided information. Since the joint prior has maximum entropy when the parameters are not correlated, the prior is constructed as the product of the marginal priors. Typically, we know what the range of reasonable or allowable values is for the model parameters. In that case, the proper distribution with maximum entropy is a uniform distribution (Toussaint, 2011). As the exponential distribution has maximum entropy among all nonnegative continuous distributions with the same average displacement (McElreath, 2018), the standard deviations of the model error terms are assigned exponential priors with averages of 0.1.

In Bayesian calibration, particular attention must be given to the choice of the prior distribution for the model bias (Brynjarsdóttir and O'Hagan, 2014). In the Kennedy and O'Hagan (2001) framework for Bayesian calibration used previously (Aerts et al., 2023), model inadequacy is a priori considered independent of the model output. To make the model parameters identifiable, we constrained the bias on the farm power to be zero by solving

$$0 = \sum_{i=1}^{N_T} \mu_{B,i} = \sum_{i=1}^{N_T} \delta_{\zeta(i)},\tag{9}$$

for  $\delta_0$  and only estimating  $\{\delta_\zeta\}_{\zeta=1}^{\zeta_{max}}$  (Aerts et al., 2023). However, the value of the model error  $E_{B,i}(\vartheta_e,\psi)$  for each turbine i depends on the choice of model parameters  $\vartheta_e$ , so simultaneously identifying both the model error and model parameters may introduce confounding of the model error with calibration parameters (Brynjarsdóttir and O'Hagan, 2014). Therefore, it is more intuitive to define the mean bias as the discrepancy  $\mu_{B,i} = \mathbb{E}_{\varphi}[P_i - \mathcal{M}_i(\vartheta_e^*,\varphi)]$  that remains when the model is calibrated with a 'best-fit' parameter  $\vartheta_e^*$  (Plumlee, 2017). If that best fit is defined as  $\vartheta_e^* = \operatorname{argmin} \mathbb{E}_{\varphi}\left[\sum_{i=1}^{N_T}(P_i - \mathcal{M}_i(\vartheta_e,\varphi))^2\right]$ , we have as a necessary condition for optimality that

$$\mathbb{E}_{\varphi} \left[ \sum_{i=1}^{N_T} \frac{\partial \mathcal{M}_i(\vartheta_e, \varphi)}{\partial \vartheta_e} \bigg|_{\vartheta_e = \vartheta_-^*} \mu_{B,i} \right] = 0. \tag{10}$$

In general, an additional constraint on the bias is added per parameter. Since we do not know the 'best-fit' parameters a priori, we can satisfy this condition trivially by requiring that  $\mu_{B,i}=0$  if  $\partial \mathcal{M}_i(\vartheta_e,\varphi)/\partial \vartheta_e\neq 0$  for at least one value of  $\vartheta_e$  with nonzero prior probability. As a result, the predicted farm power will be biased if  $\partial \mathcal{M}_i(\vartheta_e,\varphi)/\partial \vartheta_e=0$  for all  $\vartheta_e$  with nonzero prior probability. However, we find that also requiring  $\mu_{B,i}=0$  in that case works best in practice, but alternatives are explored in Sect. 4.2. Hence, the model bias parameters  $\{\delta_\zeta\}_{\zeta=0}^{\zeta_{max}}$  are all given a Dirac delta distribution centered at zero as marginal prior.

## 2.2 Bayesian updating

205

The objective of Bayesian uncertainty quantification is to construct and interpret the posterior distribution of the model parameters after Bayesian updating. The construction of the posterior based on the prior and the preceding description of the

uncertainties is discussed in Sect. 2.2.1. The posterior distribution can be used as the constituent distribution to perform a forward uncertainty quantification or obtain a stochastic flow model. In this manner, the posterior predictive distribution is obtained, which can be used to validate the adequacy of the UQ procedure as explained in Sect. 2.2.2.

## 2.2.1 Posterior distribution

Given a power measurement P and the corresponding input of the model  $\varphi$  describing the state of the wind farm and the atmosphere, the prior distribution  $p(\vartheta) = p(\vartheta_e, \vartheta_b, \vartheta_t)$  can be updated using Bayes' theorem to a posterior distribution

$$p(\vartheta|P,\varphi) = \frac{p(P|\vartheta,\varphi)p(\vartheta)}{p(P|\varphi)}.$$
(11)

The likelihood  $p(P|\vartheta,\varphi)$  of the power measurement P, given the parameters  $\vartheta$  and input to the model  $\varphi$ , is given by

$$p(P|\vartheta,\varphi) = \mathcal{N}(P;\mathcal{M}(\vartheta_e,\varphi) + \mu_B(\vartheta_b), \Sigma_T(\vartheta_t) + \Sigma_B(\vartheta_b)), \tag{12}$$

based on the description of model uncertainty in Sect. 2.1.1 and measurement uncertainty in Sect. 2.1.2. The evidence  $p(P|\varphi)$  does not depend on the parameters  $\vartheta$  and corresponds to a normalization factor. As a result, the posterior is fully determined by the prior and the likelihood.

For a dataset  $\mathcal{D} = \{P_i, \varphi_i\}_{i=1}^{N_D}$  of independent power measurements  $P_i$  with corresponding inputs to the model  $\varphi_i$ , the posterior is given by

$$p(\vartheta|\mathcal{D}) \propto p(\vartheta) \prod_{i=1}^{N_D} \mathcal{N}(P_i; \mathcal{M}(\vartheta_e, \varphi_i) + \mu_B(\vartheta_b), \Sigma_T(\vartheta_t) + \Sigma_B(\vartheta_b))$$
(13)

where the total likelihood is a product of the individual likelihoods due to independence and the prior  $p(\vartheta)$  is given by the product of the marginal priors as discussed in Sect. 2.1.3. In the limit of an infinite amount of data, the posterior converges to a point mass, given that the parameters are identifiable – see Gelman et al. (2013, p. 89) for other conditions. For a finite but large amount of data, the relative uncertainty of each of the model parameters in the posterior is inversely related to the sensitivity of the log-likelihood to that parameter, through the Fisher information matrix (Gelman et al., 2013, p. 88). Hence, the posterior parameter uncertainty represents epistemic uncertainty that can be reduced with more observations. Irreducible forms of uncertainty, such as measurement and model uncertainty, are quantified by their parameterization in the likelihood:  $\Sigma_T$  and  $\Sigma_B$  here. It is crucial that these forms of uncertainty are adequately quantified, as otherwise the the marginal posterior for the model parameters  $p(\vartheta_e|\mathcal{D})$  may be over-confident and biased (Brynjarsdóttir and O'Hagan, 2014).

#### 2.2.2 Posterior predictive distribution

The posterior predictive is the distribution of new (predicted) observations given all previous observations, the wind farm flow model, and the description of all sources of uncertainty in the likelihood and prior. For the Bayesian UQ analysis to be adequate, the original data should seem plausible under the posterior predictive distribution (Gelman et al., 2013, p. 143). Any systematic differences between the posterior predictions and the data indicate potential failings of the specified likelihood and

prior to model the actual process that generates the data (cf. Eq. (2) and (8)). For instance, if the model error uncertainty is not included in the analysis, the posterior predictive may underestimate the variance of the data. In that case, a posterior predictive check will reveal the inadequacy of the specified likelihood and prior.

The posterior predictive distribution  $p(P_{new}|\varphi_{new},\mathcal{D})$  can be rewritten as

$$p(P_{new}|\varphi_{new}, \mathcal{D}) = \int p(P_{new}, \vartheta|\varphi_{new}, \mathcal{D}) \, d\vartheta,$$

$$\stackrel{(1)}{=} \int p(P_{new}|\vartheta, \varphi_{new}) p(\vartheta|\mathcal{D}, \varphi_{new}) \, d\vartheta,$$

$$\stackrel{(2)}{=} \int p(P_{new}|\vartheta, \varphi_{new}) p(\vartheta|\mathcal{D}) \, d\vartheta,$$

$$(14)$$

where (1) requires that the new measurement is again independent from the previous ones, and (2) presumes that the posterior based on the previously observed states of the wind farm and atmosphere is independent of the new state. In practice, this means that the calibrated model and quantified model uncertainty should also be adequate for the new inflow condition  $\varphi_{new}$  (more on that in Sect. 4.4). Samples from the posterior predictive for a given model input  $\varphi_{new}^*$  are obtained by first sampling the posterior distribution  $\vartheta_e^*, \vartheta_b^* \sim p(\vartheta|\mathcal{D})$ , and then sampling from the likelihood  $P_{new}^* \sim p(P_{new}|\vartheta_e^*, \vartheta_b^*, \varphi_{new}^*)$  given the sampled parameters. Consequently, it can be interpreted as the forward UQ of the model given the epistemic uncertainty in the posterior, and the measurement and model uncertainty in the likelihood. By leaving out the measurement uncertainty, one obtains a stochastic flow model that accounts for both the epistemic uncertainty on the model parameters and the (systematic) model uncertainty.

#### 2.3 Sampling the posterior distribution

255

In practice, the analytical derivation of the moments and marginal distributions of the posterior distribution quickly becomes intractable. Instead, Markov chain Monte Carlo (MCMC) algorithms are typically used to efficiently sample the posterior distribution. With those samples, one can visualize the marginalized and joint posterior(s), consult the posterior predictive, and compute expected values. However, these algorithms are inherently serial and require  $\mathcal{O}(10^5)$  likelihood evaluations to converge to the posterior and adequately represent it (Geyer, 2011). Even for engineering models with reasonable computational expense, this can become relatively time-consuming. Therefore, we employ a variant of the transitional Markov chain Monte Carlo (TMCMC) algorithm (Ching and Chen, 2007), which is inherently parallel.

Instead of directly sampling the posterior distribution, TMCMC samples a sequence of target distributions with a sequential Monte Carlo (SMC) method. This sequence  $\pi_j(\vartheta)$  is obtained by tempering the likelihood: a stage exponent or inverse temperature  $\beta_j$  is introduced that sequentially 'cools' the target

$$\pi_j(\vartheta) \propto p(\mathcal{D}|\vartheta)^{\beta_j} p(\vartheta) \tag{15}$$

with  $j=1,\ldots,J$  and  $0=\beta_1

**Figure 1.** Schematic overview of one stage in the transitional Markov chain Monte Carlo (TMCMC) algorithm. From left to right, the samples are weighted, resampled, and perturbed. The algorithm can be parallelized in the perturbation phase. This figure is based on similar figures in literature of sequential Monte Carlo and TMCMC (e.g. Doucet et al. (2001); Minson et al. (2013); Murphy (2023)).

gradually increase the influence of the likelihood by starting with an artificially large variance and then subsequently shrinking it. As a result, the algorithm can efficiently explore the prior range and successfully sample multimodal target distributions.

On the sequence of tempered target distributions, a particular version of the Resample–Move SMC algorithm is then used (Gilks and Berzuini, 2001; Doucet et al., 2009), as depicted in Figure 1. At every stage, importance resampling is used to obtain N samples that asymptotically follow the target distribution. To that end, importance weights are computed for every j-th generation of particles

$$w(\vartheta_j^{(k)}) = \frac{p(\mathcal{D}|\vartheta_j^{(k)})^{\beta_{j+1}}p(\vartheta_j^{(k)})}{p(\mathcal{D}|\vartheta_j^{(k)})^{\beta_j}p(\vartheta_j^{(k)})} = p(\mathcal{D}|\vartheta_j^{(k)})^{\Delta\beta},\tag{16}$$

with  $\Delta\beta=\beta_{j+1}-\beta_j$  to resample the particles  $\{\vartheta_j^{(k)}\}_{k=1}^N$  with a probability

275 
$$\mathbb{P}(\Theta_{j+1}^{(k)} = \vartheta_j^{(k)}) = \frac{w(\vartheta_j^{(k)})}{\sum_{k=1}^{N} w(\vartheta_j^{(k)})} = \bar{w}(\vartheta_j^{(k)}). \tag{17}$$

Due to the importance sampling steps, the algorithm works best for priors that sufficiently cover the high-likelihood region. Since we employ wide weakly informative priors, this is almost always the case.

Then N Metropolis-Hastings (MH) MCMC chains of length L are instantiated to perturb these samples again and remove the degeneracy introduced by the resampling step (Wu et al., 2018b). In the MH-algorithm, the new sample  $\vartheta_{k+1}$  is sampled from a proposal density  $q(\vartheta_{k+1}|\vartheta_k)$  that only depends on the previous sample (Markov property). The new sample is accepted with a probability  $\min(1,\alpha_{MH})$ , where the acceptance ratio  $\alpha_{MH}$  is defined as

$$\alpha_{MH} = \frac{\pi_j(\vartheta_{k+1})}{\pi_j(\vartheta_k)} \frac{q(\vartheta_{k+1}|\vartheta_k)}{q(\vartheta_k|\vartheta_{k+1})}.$$
(18)

285

290

295

300

The L-th samples in the chains are taken as particle generation j+1. Since all MH chains can run simultaneously, the algorithm is inherently parallel. However, due to the MH steps – and thus similar to (Adaptive) MH – TMCMC does not scale as well with parameter dimension as Hamiltonian Monte Carlo, which employs gradient information. If the prior does not sufficiently cover the high-likelihood region, longer MH chains are also required to compensate for the degeneracy introduced during importance sampling.

Several components of the algorithm can be tuned, such as the proposal distribution  $q(\vartheta_{k+1}|\vartheta_k)$  in MH, the method to determine  $\beta_j$ , the number of samples per stage N, and the number of MCMC steps L. In our case, the distribution of the MH proposal is multivariate normal, that is,  $q(\vartheta_{k+1}|\vartheta_k) = \mathcal{N}(\vartheta_{k+1};\vartheta_k,\Sigma_q)$ , with as a covariance matrix  $\Sigma_q$  the importance weighted sample covariance matrix of the previous stage scaled by 1/9+8/9R, with R the observed acceptance rate (Minson et al., 2013). The cooling rate should be fast enough to reduce the computational cost but slow enough to adequately represent the next target distribution after resampling. This adequacy can be quantified by the effective sample size (ESS)  $N^{eff}$  (Gelman et al., 2013). Its interpretation is that N weighted samples  $\{w_j^{(k)}, \vartheta_j^{(k)}\}_{k=1}^N$  are worth  $N^{eff}$  i.i.d. samples drawn from the target distribution  $p_{j+1}$  (Beck and Zuev, 2013). The ESS is estimated from the normalized importance weights as

$$\hat{N}_{j}^{eff} = \frac{1}{\sum_{k} \bar{w}(\vartheta_{j}^{(k)})^{2}} = \frac{\left(\sum_{k} p(\mathcal{D}|\vartheta_{j}^{(k)})^{\Delta\beta}\right)^{2}}{\sum_{k} p(\mathcal{D}|\vartheta_{j}^{(k)})^{2\Delta\beta}}$$
(19)

The optimal cooling rate is obtained when  $\beta_{j+1}$  is chosen such that the ESS is approximately half the total number of samples per stage (Minson et al., 2013). To that end, Eq. (19) is solved for  $\beta_{j+1}$  with a bisection method on  $(\beta_j, 1]$  (Kantas et al., 2014). Based on the study of the sensitivity of the posterior to the choice of N and L with increasing parameter dimension by Minson et al. (2013), we take N=1920 and L=20 for a total of (maximum) 10 parameters. For 15 stages, this requires  $5.76\times10^5$  likelihood evaluations, of which only 300 are inherently serial. For further details on the implementation, the reader is referred to the Python toolbox UMBRA, which is released together with this paper.

## 3 Case setup

The Bayesian UQ framework is demonstrated with a reference large-eddy simulation (LES) dataset for wind-farm blockage and atmospheric gravity waves in conventionally neutral boundary layers (CNBLs). Since it is still challenging to model these effects with state-of-the-art models, the dataset provides a controlled setting for evaluating the framework under conditions of substantial model bias and uncertainty. The wind farm flow models are introduced in Sect. 3.1 and the dataset is described in Sect. 3.2. Based on the empirical model parameters and dataset characteristics, the prior distribution is further specified in Sect. 3.3.

330

#### 310 3.1 Wind farm flow models

The two wind farm flow models we will consider in this study are a standard wake model (Sect. 3.1.1) and an atmospheric perturbation model (APM) (Sect. 3.1.2). The wake model cannot capture wind-farm blockage and is expected to exhibit large model uncertainty for the considered dataset. In addition, it serves as a baseline for the model uncertainty of the APM.

#### 3.1.1 Standard wake model

The primary objective of wake models is to predict the effect of the velocity deficit downstream of a turbine on the other turbines within a farm. The most well-known and widely used wake model is presumably the one originally proposed by Jensen (1983), but over the years many others have followed (Göçmen et al., 2016). In this study, the Gaussian wake model of Bastankhah and Porté-Agel (2014) will be employed. It is based on the typical self-similar Gaussian profile of the velocity deficit and mass and momentum conservation in the wake (see also Frandsen et al. (2006)).

The velocity deficit of a turbine k located upstream is then given by

$$\frac{U_{\infty} - U_w(x)}{U_{\infty}} = \left(1 - \sqrt{1 - \frac{C_{T,k}}{8\sigma_k^{*2}}}\right) \exp\left(-\frac{1}{2\sigma_k^{*2}} \frac{y^2 + z^2}{D_k^2}\right) \mathcal{H}(x). \tag{20}$$

with  $U_{\infty}$  the free stream speed,  $U_w$  the speed in the wake, and  $\boldsymbol{x} = [x,y,z]^{\top}$  defined in a local coordinate system at the turbine hub, with z the vertical direction, z and y spanning the rotor plane, and x>0 downstream of the turbine. Further,  $\mathcal{H}$  is the Heaviside function,  $C_{T,k}$  the turbine thrust coefficient,  $D_k$  rotor diameter, and  $\sigma_k^* = \sigma_k/D_k$  the normalized wake width which grows linearly with the distance downstream from the rotor as

$$\sigma_k^* = k_w \frac{x}{D_L} + \epsilon. \tag{21}$$

Here,  $k_w$  is the wake expansion rate which is fitted to the local turbulence intensity at turbine as  $k_w = k_a I + k_b$  with  $k_a = 0.3837, k_b = 0.003678$  (Niayifar and Porté-Agel, 2016). Finally,  $\epsilon = 0.2\sqrt{(1-a)/(1-2a)}$  is a semi-empirical parameter that represents the initial wake width and a is the turbine induction factor. The turbulence intensity at the turbine is determined with the method of Niayifar and Porté-Agel (2016) based on the original (Zehtabiyan-Rezaie and Abkar, 2023) expression for the added turbulence intensity expression by Crespo and Hernández (1996)

$$I_{+} = 0.73a^{0.8325}I_{\infty}^{-0.0325}(x/D)^{-0.32},$$
(22)

with  $I_{\infty}$  the background turbulence intensity.

In order to combine multiple wakes into one flow field a wake-merging method is needed. We will use the one developed by Lanzilao and Meyers (2022). Additionally, the turbines are mirrored to capture the effect of the ground plane (Lissaman, 1979). The power extracted by turbine k can then be computed as  $P_k = \frac{1}{2}\rho A_k U_k^3 C_{P,k}(U_k)$  with  $C_{P,k}(U_k)$  the power coefficient of that turbine,  $A_k$  the swept rotor area, and  $U_k$  the disk averaged flow speed, calculated with the same quadrature method as in Allaerts and Meyers (2019).

Figure 2. Schematic representation wind-farm atmosphere interaction via atmospheric gravity waves and Internal Boundary Layer (IBL) development as modeled by the atmospheric perturbation model: (a) a sketch of the wind farm, vertical profiles of wind speed (blue) and potential temperature (purple), wave crests and troughs for internal gravity waves in the free atmosphere (red) and interface waves on the inversion layer (orange), and a developing IBL (gray); (b) the added momentum flux due to the presence of the wind farm and related to IBL growth as modeled in the APM; (c) a hypothetical displacement of the inversion layer with the associated pressure feedback  $p_t$  in the Atmospheric Boundary Layer (ABL), split up in the pressure components related to the waves on the inversion layer  $p_i$ , and the waves in the free atmosphere  $p_{fa}$ .

# 3.1.2 Atmospheric perturbation model

Atmospheric perturbation models aim to model wind-farm atmosphere interaction effects such as wind-farm blockage in addition to the turbine-scale interactions due to wakes. They do so by solving the height-averaged and linearized Reynolds-averaged Navier-Stokes (RANS) equations for the atmospheric boundary layer (ABL) under the Boussinesq approximation. The linearization involves adding a perturbation velocity  $u = [u, v, w]^{\top}$  to the velocity  $U = [U, V, W]^{\top}$  in the ABL. The APM further divides the ABL in a wind-farm layer of height  $H_1$  and a second layer of height  $H_2 = H - H_1$  with H the ABL height. The equations and their solution procedure are derived and described in detail by Allaerts and Meyers (2019); Stipa et al. (2023); Devesse et al. (2024a). The three most important terms for our purposes are the farm thrust, the added turbulent momentum flux associated with the development of an internal boundary layer (IBL), and the pressure feedback induced by the upward displacement of the capping inversion layer – the interface between the neutral atmospheric boundary layer and the stably stratified free atmosphere aloft in CNBLs. The farm trust and turbulent momentum flux contain parameters that will be calibrated with the Bayesian framework, whereas pressure feedback is crucial to capture the blockage effect. Figure 2 represents these effects schematically.

The wind-farm thrust f(x,y) is represented in the APM by filtering the turbine thrust forces  $f_k$  located at the turbine positions (Allaerts and Meyers, 2019)

$$f(x,y) = \int_{0}^{L_x} \int_{0}^{L_y} G(x - x', y - y') \sum_{k=1}^{N_t} f_k \delta(x' - x_k, y' - y_k) dx' dy'$$
(23)

with  $\delta(x,y)$  the two-dimensional Dirac delta and G(x,y) a Gaussian filter kernel with filter length scale  $L_f$ 

$$G(x,y) = \frac{1}{\pi L_f^2} \exp\left(-\frac{x^2 + y^2}{L_f^2}\right).$$
 (24)

The filter length scale is set to 1000 m by Allaerts and Meyers (2019); Devesse et al. (2024a) and 500 m by Stipa et al. (2023), and can be considered an uncertain parameter. This filtering operation must be applied to the momentum equations as a whole, but since the effect of the resulting dispersive stresses is primarily limited to the farm entrance region (Devesse et al., 2024a), we will ignore them to reduce computational cost (Devesse et al., 2024b).

The development of an internal boundary layer is accompanied by an increase of the momentum flux from the layer above the wind farm to the wind-farm layer. This added turbulent momentum flux is represented as (Devesse et al., 2024a)

$$\Delta \tau_{WF}(x, y) = a_{\tau} C_F \Pi(\mathbf{x} - d_{\tau} D \mathbf{e}_s) \tag{25}$$

with  $a_{\tau}$  the proportionality constant to the wind-farm force density  $C_F = \frac{1}{2}C_T N_T A||U_1||^2/S_F$ , and  $e_s$  is the streamwise unit vector. Here,  $C_T$  is the average turbine thrust coefficient,  $N_T$  the number of turbines, A the swept rotor area,  $U_1$  the unperturbed velocity in the farm layer, and  $S_F$  the wind farm surface. The added momentum flux is oriented along the wind-farm forcing and is zero everywhere except on the wind-farm footprint  $\Pi(x)$ . For a rectangular farm, this is a block function. To include the development of the IBL, the footprint is shifted  $d_{\tau}$  turbine diameters downstream given that the turbines that are on average aligned with the wind. That leaves two empirical parameters  $a_{\tau}$  and  $d_{\tau}$ , which were previously fitted to 0.12 and 27.8 based on the computed added momentum flux from LES data (Devesse et al., 2024a).

The farm thrust will slow down the flow in the ABL. The resulting decrease of the streamwise velocity u is balanced in the continuity equation by induced spanwise flow v and thickening of the ABL. This thickening corresponds to a lifting of the capping inversion  $\eta$ , which leads to two distinct processes that result in the pressure feedback  $p_t = p_i + p_{fa}$  (cf. Figure 2c). First, the lifting of the capping inversion directly corresponds to a cold anomaly, as the air below it is colder than the air above. These pressure perturbations  $p_i$  can travel horizontally along the capping inversion as two-dimensional interfacial gravity waves. Second, the changes in capping inversion height perturb the free atmosphere aloft, leading to internal gravity waves. These three-dimensional waves also lead to pressure perturbations, which are felt throughout the ABL (Smith, 2010). Combined, these two types of gravity waves cause a pressure increase upstream, leading to the blockage effect. Downstream, they also induce a favorable pressure gradient throughout the farm (cf. Figure 2c).

The wake effects are included with a standard wake model, which can be coupled to the height-averaged RANS equations for the two layers together with the pressure feedback from the upper atmosphere (Devesse et al., 2024b). The predicted flow redirection is included with a simplified version of the bidirectional wake merging method of Lanzilao and Meyers (2022), which is elaborated in Appendix A. In the current work, the wake model is coupled via the pressure (Stipa et al., 2023), but an upstream coupling (Allaerts and Meyers, 2019), and a velocity matching approach exist as well (Devesse et al., 2024a). Figure 3 shows the obtained flow field, where the wake model provides the information on the wakes and the APM provides the background velocity and pressure information.

Figure 3. Wind speed and pressure field obtained with an atmospheric perturbation coupled via the pressure to a Gaussian wake model with a bidirectional wake merging method. The staggered wind farm consists of 16 rows of 10 wind turbines with streamwise and spanwise spacings of 5 rotor diameters. The atmospheric boundary layer height amounts to 500 m with a capping inversion strength of 4 K and a free-atmosphere lapse rate of 8 K km $^{-1}$ . The friction velocity equals 0.275 m s $^{-1}$ . The upstream wind speed and ambient turbulence intensity at hub height are 9.24 m s $^{-1}$  and 3.93%.

## 3.2 Reference dataset

As reference data, the parametric LES study of wind-farm blockage and gravity waves in CNBLs of Lanzilao and Meyers (2024) is used. They simulated 36 selected atmospheric states based on 30 years of ERA5 re-analysis data at the nearest grid point to the Belgian–Dutch offshore wind-farm cluster. Figure 4 depicts the inflow conditions (the time-average of the last four hours of the precursor simulations) as well as the power output per turbine, normalized by the power of a hypothetical undisturbed turbine upstream, for the cases with an ABL height of 500 m. From Figure 4b it can be seen that the ABL is always neutrally stratified, since the potential temperature  $\vartheta$  remains constant  $d\theta/dz = 0$ . The capping inversion strength  $\Delta\theta$ , and the lapse rate in the free atmosphere aloft  $d\theta/dz > 0$  are varied.

The wind farm consists of 16 rows and 10 columns ( $N_T=160$ ) of 10 MW IEA reference turbines (Bortolotti et al., 2019). The streamwise and spanwise spacings are 5D, with D=198 m the turbine diameter. The turbines have hub heights of  $z_h=119$  m and in the study the thrust coefficient is fixed to  $C_T=0.88$ . The rows are counted in the streamwise direction and labeled with capital letters in Figure 4b. It can be seen in Figure 4b how the blockage effect causes large reductions in turbine power in the first rows. At the same time, the favorable pressure gradient improves power recovery in the last rows. The significant local flow redirection related to blockage misaligns the turbine wakes with the downstream turbines on the sides of

Figure 4. Large eddy simulation data of Lanzilao and Meyers (2024): (a) potential temperature profile with  $\theta_0 = 288.15$  K, averaged horizontally in space and over the last 4 h of the precursor simulation, and (b) power output per turbine normalized by the power of an 'undisturbed' upstream turbine. In (a), the ABL height is indicated with a dashed gray line and a turbine is drawn with full black lines as a reference.

the farm, resulting in a U-shaped trend in the power per row. In general, the resulting turbine power output varies significantly with atmospheric stratification.

Of the 36 available cases, we will only consider the 9 cases with an ABL height of 500 m in this study, to isolate the effect of stratification of the upper atmosphere from the height of the boundary layer. As we intend to demonstrate the framework's capability to quantify the uncertainty adequately for models of different fidelity, the APM is included as a model. However, an APM evaluation takes about 30 seconds, so 1000 model evaluations already require 8.3 core hours, compared to 0.83 core minutes for 1000 wake-model evaluations of 0.05 seconds. Although the SMC algorithm allows performing the evaluations in parallel, the total cost remains the same.

#### 3.3 Prior choices depending on the dataset and models

An overview of the marginal priors for  $\vartheta = \{\vartheta_b, \vartheta_e\}$  is given in Table 1. The measurement error covariance can be calculated from the 90-minute long turbulent power signals using the moving block bootstrap (Garcia et al., 2005). Since the upstream velocity and potential temperature profiles are available from the LES, the inflow uncertainty is considered negligible. The prior of the mean bias terms depends on the considered wind farm flow model in the adapted Bayesian framework. We know a priori that the sensitivity of the predicted power by the wake model to the wake expansion rate is only nonzero for the upstream turbines, which are not waked. Therefore, only the bias in the upstream and undisturbed rows of turbines  $\delta_0$  can be non-zero based on the condition against confounding of the model bias with calibration parameters in Eq. (10). However, we find that also requiring  $\delta_0 = 0$  works best in practice based on a comparison of alternatives in Sect. 4.2. For the APM, the condition for the best-fit interpretation of the model parameters in Eq. (10) directly requires that all mean bias terms  $\delta_{\zeta(i)}$  are zero.

**Table 1.** Prior distributions for the inverse uncertainty quantification of the wake model and the atmospheric perturbation model. Uniform distributions on an interval [a,b] are abbreviated as  $\mathcal{U}_a^b$ . The exponential distribution  $\operatorname{Exp}_{\lambda}$  has a PDF  $p(x) = \lambda^{-1} \exp(-x/\lambda)$  for x > 0. The Dirac delta distribution  $\operatorname{Delta}(a)$  has a PDF  $p(x) = \delta(x-a)$ . The filter length  $L_f$  and height of the first layer  $H_1$  are expressed in meters and the spatial delay of the turbulent entrainment  $d_{\tau}$  in turbine rotor diameters.

| $artheta_e$       |                       |                            |                           |                   |                      | $\vartheta_b$       |                            |
|-------------------|-----------------------|----------------------------|---------------------------|-------------------|----------------------|---------------------|----------------------------|
| $k_a$             | $k_b$                 | $L_f$                      | $H_1$                     | $a_{\tau}$        | $d_{	au}$            | $\delta_{\zeta(i)}$ | $\sigma_{B,\zeta(i)}$      |
| $\mathcal{U}_0^1$ | $\mathcal{U}_0^{0.1}$ | $\mathcal{U}_{500}^{2000}$ | $\mathcal{U}^{450}_{220}$ | $\mathcal{U}_0^1$ | $\mathcal{U}_0^{80}$ | Delta(0)            | $\operatorname{Exp}_{0.1}$ |

The model parameters are given by  $\vartheta_e = \{k_a, k_b\}$  for the wake model and by  $\vartheta_e = \{k_a, k_b, L_f, H_1, a_\tau, d_\tau\}$  for the APM. Similar to previous work (Aerts et al., 2023), the wake expansion rate parameter  $k_a$  gets a uniform prior over the unit interval, whereas  $k_b$  gets a stronger uniform prior between 0 and 0.1. The filter length scale  $L_f$  is not taken smaller than the grid spacing of 500 m and not larger than two times the current value in Devesse et al. (2024a). The allowable farm layer height  $H_1$  is chosen slightly larger than the turbine tip height of 218 m and smaller than 90% of the ABL height of 500 m. The strength  $a_\tau$  of the turbulent entrainment should be positive and is not expected to be more than 10 times larger than its current estimated value of  $\approx 0.1$  (Devesse et al., 2024a). The spatial delay of the turbulent entrainment  $d_\tau$  should clearly be positive and cannot exceed the farm length of 80 turbine diameters.

## 4 Results and discussion

430

435

Before proceeding to a more practical demonstration of the framework with a wake model and an atmospheric perturbation model, we demonstrate the consequences of not properly including model error in Sect. 4.1 based on an analytical example using farm power. The inverse UQ analyses for the wake model and the APM are conducted based on the turbine power. In this manner, the model adequacy in representing wake, blockage, flow redirection, and pressure gradient effects can be assessed. In Sect. 4.2, the adequacy of the Bayesian framework is verified for the wake model, which is expected to show large model error and uncertainty for the blockage dataset, and the APM, which is expected to perform better. The posterior distributions for both models are compared in Sect. 4.3. Lastly, the generalization of the results as well as the intended use of the framework are discussed in Sect. 4.4. The posterior distributions are sampled on the wICE supercomputing platform of the VSC (Vlaams Supercomputer Centrum), using Sapphire Rapids nodes containing 2 Intel Xeon Platinum 8468 CPUs (48 cores each).

#### 4.1 Consequences of neglecting model error in Bayesian UQ

To illustrate what goes wrong when the model error is not properly included in the Bayesian framework, we consider a simple model for the farm power

$$440 \quad P_f = \vartheta_e N_T P_{\infty}, \tag{26}$$

where the empirical parameter  $\vartheta_e$  represents the efficiency of the wind farm and  $P_\infty$  is the power of a hypothetical undisturbed turbine upstream. Although the efficiency is highly variable for the considered  $N_D=9$  stratification regimes, the model parameter  $\vartheta_e$  is assumed to be constant – as in all previous Bayesian UQ analyses of wind-farm flow model parameters. However, since this assumption is clearly invalid in the present case, properly accounting for model error becomes essential.

#### 445 4.1.1 Posterior distribution

The Bayesian framework yields the following joint posterior distribution for the model parameter  $\vartheta$  and the standard deviation of the model error  $\sigma_B$  through Eq. (13)

$$p(\vartheta_e, \sigma_B | \mathcal{D}) \propto p(\vartheta_e) p(\sigma_B) \prod_{i=1}^{N_D} \mathcal{N}\left(P_{f,i} / (N_T P_\infty); \vartheta_e, \sigma_T^2 + \sigma_B^2\right). \tag{27}$$

The uncertainty due to finite-time averaging  $\sigma_T$  is obtained for simplicity as the average of the bootstrap estimates  $\sigma_{T,i}$  for each simulation. The marginal prior of the standard deviation of the model error is an exponential distribution  $p(\sigma_B) = \lambda^{-1} \exp(-\sigma_B/\lambda)$  with mean  $\lambda = 0.1$ . The marginal prior of the wind speed reduction factor is a normal distribution with mean  $\mu_0$  and standard deviation  $\sigma_0$ , that is,  $p(\vartheta_e) = \mathcal{N}(\vartheta_e; \mu_0, \sigma_0^2)$  with  $\mu_0 = 0.5$  and  $\sigma_0 = 0.1$ .

We can examine the effect of neglecting model uncertainty by comparing the conditional posterior  $p(\vartheta_e|\sigma_B, \mathcal{D})$  with  $\sigma_B=0$  and with  $\sigma_B$  equal to the mode of the marginal posterior  $p(\sigma_B|\mathcal{D})$ . For this simple example, the conditional posterior of the model parameter  $\vartheta_e$  is a normal distribution  $p(\vartheta_e|\sigma_B, \mathcal{D}) = \mathcal{N}(\vartheta_e; \mu_{N_D}, \sigma_{N_D}^2)$  with mean  $\mu_{N_D}$  and variance  $\sigma_{N_D}^2$ , equal to

$$\mu_{N_D} = \frac{\sigma_0^2 \sum_{i=1}^{N_D} P_{f,i} / (N_T P_\infty) + \sigma_M^2 \mu_0}{N_D \sigma_0^2 + \sigma_M^2},\tag{28}$$

$$\sigma_{N_D}^2 = \frac{\sigma_0^2 \sigma_M^2}{N_D \sigma_0^2 + \sigma_M^2},\tag{29}$$

with  $\sigma_M^2 = \sigma_B^2 + \sigma_T^2$  the total variance. Note that by increasing the number of measurements  $N_D$ , the posterior indeed converges to a point mass, in this case centered at the sample mean. Given enough data and for  $\lambda$  sufficiently large, the mode of  $p(\sigma_B|\mathcal{D})$  converges to  $\sqrt{s^2 - \sigma_T^2}$ , where  $s^2$  is the sample variance. Consequently, the estimated model error variance will capture the remaining variance in the data. If one assumes that  $\sigma_B = 0$  when there is non-negligible model uncertainty, Eq. (28) and (29) show that the posterior mean  $\mu_{N_D}$  is biased and the uncertainty  $\sigma_{N_D}$  underestimated. This is demonstrated for the example at hand in Figure 5a, as the posterior distribution obtained by neglecting the model error is highly overconfident. Also note that the SMC samples of the marginal posterior  $p(\vartheta_e|\mathcal{D})$  agree very well with the analytical conditional posterior  $p(\vartheta_e|\sigma_B,\mathcal{D})$  with  $\sigma_B$  equal to  $\sqrt{s^2 - \sigma_T^2}$ .

# 4.1.2 Posterior predictive distribution

460

465

The posterior predictive distribution for a new measurement  $P_{f,new}$  is given by

$$p(P_{f,new}|\mathcal{D}) = \mathcal{N}(P_{f,new}/(N_T P_{\infty}); \mu_{N_D}, \sigma_B^2 + \sigma_T^2 + \sigma_{N_D}^2). \tag{30}$$

Figure 5. Comparison of the results of Bayesian UQ with and without the inclusion of model error  $\mathcal{E}_B$ : (a) posterior distribution, and (b) posterior predictive distribution. The data and SMC samples of the distributions are given as histograms and the analytically obtained probability density functions as full lines.

The posterior predictive variance consists of the model variance, the measurement variance, and the propagated posterior parameter variance. If the model error is not included in the analysis, the posterior predictive underestimates the variance of the data both by neglecting  $\sigma_B^2$  and underestimating  $\sigma_{N_D}^2$ . Figure 5b shows that in the current example, the data do not seem plausible under the posterior predictive that neglects model error, rendering such a Bayesian analysis inadequate. The proper inclusion of model error through our framework yields an adequate posterior predictive and Bayesian analysis. Although the current example exhibits exceptionally large model uncertainty, most if not all of the current wind farm flow models have non-negligible model error, and Bayesian UQ analyses of such models that neglect model error will suffer similar issues.

## 4.1.3 Implications for stochastic flow models

480

Current stochastic wake models are obtained by propagating the posterior of the parameters – obtained by ignoring model error – through the model (Zhang and Zhao, 2020), they only account for the uncertainty on their parameters due to limited calibration data, and do not capture the uncertainty due to varying unmodeled physics, such as stratification effects, time-resolved turbulence, and so forth. Given enough data (here only 9 observations), they will therefore significantly underestimate the variability of the true process, as demonstrated in Figure 5b. Moreover, in the limit of infinite data such 'stochastic' models will in fact become deterministic as seen in Eq. (29) and (30) with  $\sigma_{N_D} \to 0$  and  $\sigma_B = 0$ . Therefore, truly stochastic wind farm flow models should include both the posterior parameter uncertainty, quantified in  $p(\vartheta_e|\mathcal{D})$ , and the model uncertainty, quantified in  $\Sigma_B$  in the framework.

Figure 6. Comparison of the turbine power from LES with the posterior predictive distributions for (a) the wake model and (b) the atmospheric perturbation model. The mean power is shown as a solid line, with shaded regions indicating one standard deviation above and below the mean. The mean and standard deviation of the model outputs  $\mathcal{M}(\vartheta_e, \varphi)$  for the posterior samples of  $\vartheta_e$  are shown as well after adding the measurement error  $\mathcal{E}_M$ . The reference data points are also plotted as individual dots.

# 4.2 Adequacy of the Bayesian framework

We now turn to the analysis of the wake model and the APM with turbine power data. Since the posterior distribution is intractable for those cases, we use the parallelized sequential Monte Carlo algorithm implemented in UMBRA to sample it (Sect. 2.3). The SMC algorithm yields 1920 samples of the joint posterior distribution of  $\vartheta_e$  and  $\vartheta_b$  after convergence. In addition, we generate a sample of the posterior predictive distribution for each sample of the posterior so that the adequacy of the Bayesian framework can be assessed by comparing the posterior predictive samples with the observations (Sect. 2.2.2).

Figure 6 compares the means and standard deviations of the posterior predictive samples of each turbine with the distribution of the LES reference data for both the wake model and the APM. To isolate model uncertainty from (epistemic) parameter uncertainty and measurement uncertainty, we also show the posterior predictive obtained from the same posterior of the model parameters, but with  $\{\sigma_{B,\zeta(i)}\}_{i=1}^{N_T}$  set to zero. Figure 6a shows that the wake model exhibits substantial model uncertainty for this dataset. In the first turbine rows, the model uncertainty is inflated because of the significant bias due to wind-farm blockage. Further downstream, the large model uncertainty stems form the U-shaped power variations caused by flow redirection and the enhanced power recovery due to the favorable pressure gradient, which the wake model fails to capture. Notably, the data appear implausible under the posterior predictive without model error, underscoring the importance of properly accounting for model uncertainty. Figure 6b shows that the APM yields considerably lower model uncertainty, as it successfully incorporates blockage, flow redirection and pressure gradient effects. However, consistent with previous findings (Devesse et al., 2024b), the APM underestimates blockage effects in the first rows, which inflates model uncertainty for those turbines. Since the reference data are deemed plausible under the posterior predictive distributions of both models, the Bayesian framework proves to be adequate.

Figure 7. Comparison of the turbine power data from LES with the posterior predictive distribution for the wake model when the mean bias terms  $\mu_{B,i}$  are estimated together with the model parameters. In (a) all mean bias terms are estimated with the constraint that there sum is zero, whereas in (b) only the bias on the upstream turbines is estimated. The mean power is shown as a solid line, with shaded regions indicating one standard deviation above and below the mean. The reference data points are also plotted as individual dots.

To avoid overestimating model uncertainty caused by systematic model mismatch, one could also estimate the mean model bias. However, this bias must be constrained to ensure model parameter identifiability, either by enforcing unbiased predicted farm power in Eq. (9) or by trivially satisfying the condition against confounding of the model bias with model parameters in Eq. (10). One approach is to relax the condition for the 'best-fit' parameter interpretation in Eq. (10), as illustrated in Figure 7a for the wake model. This allows identifying the mean bias  $\delta_0$  in the first turbine rows due to blockage, resulting in lower estimated model uncertainty. However, to compensate for this bias and still match farm power, the calibrated wake model systematically underestimates turbine power downstream. Alternatively, we can relax the constraint that predicted farm power must be unbiased, as shown in Figure 7b for the wake model. This yields the exact same calibration (and posterior of the model parameters) as when the mean bias for the upstream turbines is not estimated, since upstream power contains no information about the model parameters. Nevertheless, as Figure 6b shows, even flow models that have parameters that influence the predicted blockage effect can exhibit bias, leading to inflated uncertainty estimates for these models. Therefore, we recommend excluding mean model bias to enable the objective comparison of model uncertainty between different flow models, even though the estimated model uncertainty is conservative.

#### 4.3 Model comparison with quantified model and parameter uncertainty

The Bayesian UQ framework can be used to perform objective model comparison with quantified parameter and model uncertainty given a reference dataset. In Sect. 4.3.1, the posterior distributions of the model parameters are presented for the wake model and the atmospheric perturbation model. The posterior distribution of the parameters describing the model error distribution are compared for both models in Sect. 4.3.2.

Figure 8. Joint posterior probability density of the parameters in the wake expansion rate  $k_w = k_a I + k_b$ . For each parameter, the median is given together with the 2.5% and 97.5% quantiles, expressed as relative deviations from the median.

## 4.3.1 Posterior distribution of the model parameters

Figure 8 shows one- and two-dimensional histograms of the joint posterior distribution of the wake model parameters, based on samples generated with SMC in UMBRA and visualized with Corner (Foreman-Mackey, 2016). A comparison between the marginal posterior and prior distributions reveals that the parameters are well-identified. Notably, the posterior median of  $k_a$  (0.52) exceeds its standard reference value of 0.384 (Niayifar and Porté-Agel, 2016), while  $k_b$  is estimated to be nearly zero. In rows A and B, where turbines are not affected by wakes, the mean posterior wake expansion rate  $k_w = k_a I + k_b$  is approximately 0.018 – given an ambient turbulence intensity of 0.039 – compared to 0.019 based on the same turbulence intensity I and literature values for  $k_a$  and  $k_b$ . In the downstream rows, the mean posterior wake expansion rate increases to around 0.073 due to wake-added turbulence, compared to a literature-based value of 0.064 using the same local turbulence intensity. This suggests that wake recovery in waked turbine rows is overestimated to compensate for the favorable pressure gradient influencing the background flow field. Although the estimated wake expansion rate in the upstream rows is consistent with earlier results, the local wind speed is largely overestimated by the wake model, which makes the comparison invalid.

Figure 9 shows the samples of the joint posterior distribution of the APM parameters, which are all well identified. Compared to standard values, the posterior median of  $k_a$  is lower (0.32) and that of  $k_b$  is higher (0.0125), resulting in a larger mean wake expansion rate in rows A–B ( $k_w \approx 0.027$ ) and a similar rate in the downstream rows ( $k_w \approx 0.064$ ), given the same turbulence intensities. The increased upstream wake expansion rate suggests the need for further investigation into turbine wakes under blockage conditions (see e.g. Ndindayino et al., 2025). The estimated filter length scale  $L_f$  is smaller than the value used in Devesse et al. (2024a), but slightly larger than in Stipa et al. (2023). The relatively high uncertainty in  $L_f$  is attributed to the limited sensitivity of the turbine power to changes in filter width between 500 m to 640 m. The first-layer height  $H_1$  is estimated to be close to twice the turbine hub height, consistent with findings from a previous parameter study (Allaerts and

**Figure 9.** Joint posterior probability density of the parameters in the atmospheric perturbation model. For each parameter, the median is given together with the 2.5% and 97.5% quantiles, expressed as relative deviations from the median.

Meyers, 2019). The estimated strength of the wind-farm added momentum flux  $a_{\tau}$  reaches the upper bound of its prior, roughly ten times its current value, and its spatial delay  $d_{\tau}$  is larger than previously fitted. This discrepancy across the model chain indicates that the parameterization of the added momentum flux requires further refinement. In the pressure-based coupling of the wake model to the height-averaged RANS equations, noticeable increases in turbine power only occur for  $a_{\tau}$  values near one. Thus, the strong added momentum flux downstream helps replicating the observed power increase in rows N to P (cf. Figure 6b).

**Table 2.** Summary statistics of the marginal posterior distributions of the model error standard deviation  $\sigma_{B,\zeta(i)}$  for the wake model (WM) and the atmospheric perturbation model (APM). For each parameter, the median is given together with the 2.5% and 97.5% quantiles, expressed as relative deviations from the median.

| [%] | $\sigma_{B,0}$        | $\sigma_{B,1}$        | $\sigma_{B,2}$        | $\sigma_{B,3}$      |
|-----|-----------------------|-----------------------|-----------------------|---------------------|
| WM  | $27^{+11\%}_{-10\%}$  | $8.6^{+12\%}_{-10\%}$ | $4.6^{+11\%}_{-11\%}$ | $6.1^{+5\%}_{-5\%}$ |
| APM | $4.9^{+13\%}_{-12\%}$ | $5.1^{+13\%}_{-11\%}$ | $6.1^{+12\%}_{-11\%}$ | $2.8^{+6\%}_{-6\%}$ |

## 4.3.2 Comparison of the quantified model uncertainty

Table 2 summarizes the marginal posteriors of the model error standard deviations for both wind farm flow models. It is clear that the rather large uncertainties  $\sigma_{B,\zeta(i)}$  for the wake model are caused by the unobserved or unmodeled variations in atmospheric stratification in this dataset. In general, the model uncertainty is smaller for the APM, as the standard deviations of the model error  $\sigma_{B,\zeta(i)}$  are smaller. Because the APM captures the blockage effect, the model uncertainty on upstream turbines is reduced by a factor 5.5 for this dataset. In addition, variations in upstream blockage are better estimated as the uncertainty of the turbines with one waking turbine is lowered by a factor 1.5. Only the model uncertainty for turbines with two upstream waking turbines is larger. In fact, it is seen in Figure 6 that the deviations from the predicted power are larger in rows E and F. In contrast, the model uncertainty  $\sigma_{B,3}$  for turbines farther downstream is a factor 2.2 lower than for the wake model. This is because the APM adequately models the increase in power in later rows due to the inclusion of the favorable pressure gradient and the wind-farm added momentum flux. Note that the uncertainty on  $\sigma_{B,3}$  is smaller than on the other bias variances. This is because  $\sigma_{B,3}$  is associated with 100 turbines, while the others are associated with 20 only. Thus, its epistemic uncertainty is further reduced by a factor  $\sqrt{5}$ .

## 4.4 Generalization of the obtained results

A natural question that arises when calibrating models is to what extent the resulting performance generalizes to other datasets. Since physics-based wind farm flow models are expected to generalize well to other farm lay-outs, wind speeds, and wind directions, we expect that the performance should generalize well when varying those conditions. In that case the posterior distribution of the empirical parameters  $p(\vartheta_e|\mathcal{D})$  can be used together with the quantified model error distribution to obtain a stochastic wind farm flow model. However, for unobserved or unmodeled conditions  $\psi'$ , we cannot expect a proper generalization, since the bias of the flow model can only (partially) be reduced by altering the empirical model parameters  $\vartheta_e$ . Consequently, the results of the uncertainty quantification depend largely on the resemblance between the distribution of these unobserved conditions in the considered data  $p(\psi')$  and their true distribution  $p_{true}(\psi')$ . In this case, we have by no means covered the true distribution  $p_{true}(\psi')$  – even if we intended to consider only atmospheric stratification effects in CNBLs with an ABL height of 500 m. Similarly, the posteriors of the model parameters likely depend on the ABL height. As such, the obtained results are specific to the dataset considered and are intended only as an illustration of the methodology. However,

we showed that the methodology does allow for objective model comparison with quantified model and parameter uncertainty given a benchmark dataset.

#### 575 **Summary and outlook**

Bayesian UQ leverages data to quantify the uncertainty of model parameters and the model itself by updating a prior distribution that characterizes the available knowledge before having seen the data, to a posterior distribution. In this manuscript, we examined the use of Bayesian UQ for (1) obtaining stochastic wind farm flow models through Bayesian calibration, and (2) objective model comparison with quantified parameter and model uncertainty. As both applications require that model inadequacy is properly taken into account, the model inadequacy formulation in a previously developed Bayesian UQ framework (Aerts et al., 2023) was improved. The framework was demonstrated with engineering models for wind-farm atmosphere interaction using a large-eddy simulation dataset for wind-farm blockage due to atmospheric gravity waves. In doing so, the framework was tested for delineated data with a large anticipated model uncertainty, as current engineering models face difficulties in representing those effects. In contrast to earlier studies, we used a parallelized sequential Monte Carlo algorithm based on likelihood tempering to speed up the approximation of the posterior, though at a similar computational cost. This complete workflow is made available in a Python toolbox coined Uncertainty Modeling toolbox for Bayesian data Re-Analysis (UMBRA) which can be used together with WAYVE (Devesse et al., 2023) and the Wind-Farm API (Quick et al., 2024).

With a simple example model for wind farm power, the consequences of not properly including model error in Bayesian UQ are illustrated. On the one hand, the posterior distribution of the model parameters is overconfident and biased when the model error is neglected. In that case, a posterior predictive check also shows that the Bayesian analysis is inadequate. Hence, the proper inclusion of model error is also important when one is only interested in the posterior distribution of the flow model parameters. On the other hand, current stochastic flow models, which only propagate the posterior distribution of the model parameters through the model (Zhang and Zhao, 2020), may significantly underestimate the variability of the true process that the model aims to represent, as soon as there is non-negligible model error. Moreover, in the limit of infinite data such 'stochastic' models will, in fact, become deterministic. The presented framework properly includes the model error so that the posterior distribution of the model parameters and the model uncertainty are adequately quantified, also in the limit of infinite data. Since most if not all of the current wind farm flow models have non-negligible model error, Bayesian UQ analyses that neglect model error will suffer similar issues.

The adequacy of incorporating model error on turbine power predictions within the Bayesian framework was also assessed. A posterior predictive check using the blockage dataset revealed that the framework is adequate for both a standard wake model, which does not capture wind-farm blockage effects and has large model error, and for an atmospheric perturbation model (APM), which does capture those effects. By requiring that the calibrated model is unbiased on the farm power (Aerts et al., 2023) and that the calibration parameters are to be interpreted as 'best-fit' parameters (Plumlee, 2017), the mean bias on each turbine is a priori considered to be zero. By doing so, the current approach does not suffer from confounding of

calibration parameters with model inadequacy (Brynjarsdóttir and O'Hagan, 2014). Although the quantified model uncertainty is more conservative as a result, the model uncertainty can be objectively compared for different wind farm flow models.

The Bayesian UQ of the wake model and APM showed that the framework can be used for objective model comparison with quantified parameter and model uncertainty given a reference dataset. The posterior distribution of the model parameters is significantly updated with respect to the prior distribution, indicating that the parameters are well identified. The uncertainty of the parameters characterizes the remaining uncertainty due to the limited amount of data, and it is seen that the relative uncertainties of the model parameters are inversely related to their sensitivity. Posterior correlations and inconsistencies between the posterior modes throughout the model chain may inform modelers of the parts that need to be further improved. For the APM, the estimated wake expansion rate in the upstream turbine rows is higher than the rate derived from standard parameter values (Niayifar and Porté-Agel, 2016), while in the downstream rows, they align closely. This encourages further research on turbine wakes under blockage conditions (e.g. Ndindayino et al., 2025). A comparison of the quantified model uncertainties shows that the APM exhibits substantially lower uncertainty than the wake model. This applies both to the upstream turbines, which are subject to significant blockage effects, and to the downstream turbines, which benefit from the favorable pressure gradient across the farm in the considered dataset. Incorporating parameter uncertainty into model uncertainty quantification enables a more robust assessment of model performance under specific atmospheric conditions, which is relevant for applications such as production forecasting and wind-farm flow control.

Further research may use the method to formally compare the model-form uncertainty for wind farm flow models of different complexity given a benchmark dataset. In addition, the parameter and model uncertainty quantified in the posterior can inform robust wind farm flow control and layout optimization. When using the framework with operational farm data, the uncertainty on the inflow conditions can also be incorporated using a similar approach to the hierarchical stochastic prior. In doing so, the model uncertainty may also be separated from the uncertainty in the inflow conditions. Lastly, further research into the accuracy of approximate Bayesian inference methods, such as the Laplace approximation, variational methods, and Gaussian process emulators, in this setting is also of interest to reduce the computational cost of the methodology.

Code and data availability. The large-eddy simulation dataset for wind-farm blockage and atmospheric gravity waves in conventionally neutral boundary layers that is used in this work is publicly available (https://doi.org/10.48804/LRSENQ). The code for the wake model and atmospheric perturbation model are available in the Python package WAYVE (https://gitlab.kuleuven.be/TFSO-software/wayve). The code used to perform the Bayesian uncertainty quantification with a parallelized sequential Monte Carlo algorithm is made available in a Python package coined UMBRA: Uncertainty Modeling toolbox for Bayesian data Re-Analysis (https://gitlab.kuleuven.be/TFSO-software/umbra).

# Appendix A: Including local flow redirection in the wake-model coupling in WAYVE

The reduction of the streamwise velocity due to blockage is accompanied by both an increase in the ABL height and a spanwise velocity increase directed away from the centerline of the farm. Hence, the background flow is bidirectional and can be

formulated as

655

$$\boldsymbol{U}_b(\boldsymbol{x}) = \|\boldsymbol{U}_b(\boldsymbol{x})\|_2 \begin{pmatrix} \cos\theta_b(x,y) & \sin\theta_b(x,y) & 0 \end{pmatrix}^\top.$$
(A1)

Since the height-averaged ABL equations solved in the APM do provide a spanwise velocity perturbation, it is expected that a bidirectional wake model will perform better.

Lanzilao and Meyers (2024) derived their wake-merging method for a heterogeneous background velocity field characterized by changes in direction and magnitude. By assuming that the wake only affects the velocity component perpendicular to the rotor and not the velocity component parallel to it that may develop downstream, they arrive at the recursion formula

$$U_k(x) = (U_{k-1}(x) \cdot e_{\perp,k})[1 - W_k(X_k(x))]e_{\perp,k} + (U_{k-1}(x) \cdot e_{\parallel,k})e_{\parallel,k}, \tag{A2}$$

with  $e_{\perp,k} = (\cos\theta_k, \sin\theta_k, 0), e_{\parallel,k} = (-\sin\theta_k, \cos\theta_k, 0)$  and  $U_0(\boldsymbol{x}) = U_b(\boldsymbol{x})$ . Since the wakes are transported by the mean flow, they introduce a local coordinate system  $\boldsymbol{X}_k(\boldsymbol{x}) = (X_k(\boldsymbol{x}), Y_k(\boldsymbol{x}), Z_k(\boldsymbol{x}))$  that is oriented along the streamlines of the background flow field

$$X_k(\boldsymbol{x}) = \int_{x_k}^x \cos\theta_b(\bar{x}, y) d\bar{x} + \int_{y_k}^y \sin\theta_b(x, \bar{y}) d\bar{y},$$
(A3)

$$Y_k(\boldsymbol{x}) = -\int_{x_h}^{x} \sin\theta_b(\bar{x}, y) d\bar{x} + \int_{y_h}^{y} \cos\theta_b(x, \bar{y}) d\bar{y}, \tag{A4}$$

$$Z_k(\mathbf{x}) = z - z_{h,k}. \tag{A5}$$

Since engineering models are mostly designed to be as cheap as possible (for efficient AEP evaluations, layout optimization and wind-farm flow control), it is desirable to circumvent the two integrations per turbine over the whole wake center line. Therefore, we take a similar approach as Stipa et al. (2024) to reduce computational costs – albeit with another wake merging method. They argue that the scale at which the local wind direction changes is much larger than the turbine wake scale which allows ignoring the advection of wake deficits. If additionally, all turbines are aligned with the background flow,

$$\theta_k = \theta_b(\boldsymbol{x}_k) \tag{A6}$$

$$U_b(\boldsymbol{x}_k) \cdot \boldsymbol{e}_{\perp,k} = \|U_b(\boldsymbol{x}_k)\|_2,\tag{A7}$$

wake deflection through yaw misalignment can be neglected. The assumption of slowly varying background wind direction compared to the wake scale allows setting  $\theta_b(x,y) \approx \theta_b(x_k,y_k)$  in the integrals such that

$$\boldsymbol{X}_{k}(\boldsymbol{x}) = \begin{bmatrix} \cos\theta_{b}(\boldsymbol{x}_{k}) & \sin\theta_{b}(\boldsymbol{x}_{k}) & 0\\ -\sin\theta_{b}(\boldsymbol{x}_{k}) & \cos\theta_{b}(\boldsymbol{x}_{k}) & 0\\ 0 & 0 & 1 \end{bmatrix} (\boldsymbol{x} - \boldsymbol{x}_{k}). \tag{A8}$$

With these analytical solutions to the integrals, the recursion formula can be made fully explicit. After some matrix manipulations

$$\boldsymbol{U}_{k}(\boldsymbol{x}) = \begin{bmatrix} \boldsymbol{e}_{\perp,k} & \boldsymbol{e}_{\parallel,k} \end{bmatrix} \begin{pmatrix} (\boldsymbol{U}_{k-1}(\boldsymbol{x}) \cdot \boldsymbol{e}_{\perp,k})[1 - W_{k}(\boldsymbol{X}_{k}(\boldsymbol{x}))] \\ \boldsymbol{U}_{k-1}(\boldsymbol{x}) \cdot \boldsymbol{e}_{\parallel,k} \end{pmatrix}$$
(A9)

$$= \begin{bmatrix} e_{\perp,k} & e_{\parallel,k} \end{bmatrix} \begin{bmatrix} e_{\perp,k}^{\top} [1 - W_k(\boldsymbol{X}_k(\boldsymbol{x}))] \\ e_{\parallel,k}^{\top} \end{bmatrix} \boldsymbol{U}_{k-1}(\boldsymbol{x}),$$
(A10)

it is found that (Devesse et al., 2024a)

665 
$$U(x) = \left(\prod_{k=1}^{N_t} B_k(x)\right) U_b(x),$$
 (A11)

with

$$B_k(\boldsymbol{x}) = \boldsymbol{e}_{\perp,k} \boldsymbol{e}_{\perp,k}^{\top} [1 - W_k(R_k(\boldsymbol{x} - \boldsymbol{x}_k))] + \boldsymbol{e}_{\parallel,k} \boldsymbol{e}_{\parallel,k}^{\top}, \tag{A12}$$

and  $R_k$  the rotation matrix defined in Eq. (A8). With a vectorized implementation, the bidirectional wake-merging method has a similar computational cost as the unidirectional method.

Author contributions. F. A., K. D., and J. M. jointly defined the scope of the study. F. A. developed the Bayesian inference framework, implemented necessary algorithms, and performed all simulations. K. D. provided support for the atmospheric perturbation model and data processing. The manuscript was written by F. A. and J. M. and edited by K. D..

Competing interests. At least one of the (co-)authors is a member of the editorial board of Wind Energy Science. The authors have no other competing interests to declare.

Acknowledgements. The authors acknowledge funding from the European Union Horizon Europe Framework program (HORIZON-CL5-2021-D3-03-04) under grant agreement no. 101084205. The computational resources and services in this work were provided by the VSC (Flemish Supercomputer Center), funded by the Research Foundation Flanders (FWO) and the Flemish Government department EWI.

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
