# Peer review of "Bayesian uncertainty quantification of engineering models for wind-farm atmosphere interaction"

_Wind Energy Science, 2025_

## Referee Comment (RC2)

The manuscript "Bayesian uncertainty quantification of engineering models for wind-farm atmosphere interaction" by Aerts et. al. presents a Bayesian UQ framework for calibrating and comparing wind farm flow models. The Bayesian formulation represents both model error and measurement error and is demonstrated for two engineering models given LES data of a wind farm situated in a conventionally neutral boundary layer. It is shown that, for this dataset, the Bayesian framework quantifies the reduced model error in the higher fidelity model when compared to the lower fidelity model, as well as the uncertainty in each model parameter. Overall the paper is well written, and the examples and figures are effective at demonstrating the utility of the Bayesian UQ framework. High-level comments and line-specific comments are enumerated below.

1. Please ensure that all symbols and variables are defined in the text throughout the manuscript. I found myself inferring the meaning of several statistical quantities, which may be difficult for the general wind energy audience. A nomenclature section at the beginning of the manuscript would greatly help to enhance the readability of the paper.

2. The authors highlight the importance of assessing model uncertainty to provide a robust evaluation of AEP predictions from low-fidelity engineering models, particularly for applications such as wind farm flow control and layout optimization. In wind farm flow control, AEP is often estimated using engineering models (such as the standard wake model considered in this study) across a range of control parameters, with typical AEP gains over baseline controls of around 1% or even less observed. Given the methodologies and results presented in this manuscript, can the authors provide any insight into whether such small predicted power gains are at all meaningful in light of the large uncertainty and model error observed with the low-fidelity engineering model, such as those shown in Figure 6? Is this error something that can be improved with more data, or are the deficiencies in the model simply too large for low-fidelity engineering models to make meaningful wind farm planning decisions? While rigorously addressing these questions may extend beyond the scope of this manuscript, commenting on them in the conclusion or suggesting them as directions for future work could provide a valuable perspective for the wind energy industry.

3. L44: Define the first instance of "UQ', and please ensure all acronyms are defined the first time they are used.

4. L185: What is the justification of using the 0.1 value? Can this be clarified in the text?

5. L320: In standard wake models such as an empirically tuned Gaussian model like the one described in Section 3.1.1, it is common to introduce a multiplication factor to scale the inflow wind speed or power coefficient of the turbine to account for the turbine induction field, blockage effects, and other discrepancies with BEM theory. Is such a scaling factor included here? Would that improve the performance of the standard wake model in Section 4, particularly the large modeling error observed in the first row of turbines in Figure 6?

6. L327: This sentence may give the false impression that the wake expansion coefficients are fixed in this study to values of ka = 0.3837, kb = 0.003678. Could this be reworded to clearly state that these are values found in a previous study, and that the wake expansion coefficients will be included as uncertain parameters for the Bayesian UQ analysis?

7. L334: Can additional information be added to this sentence to briefly describe the wake merging method used in this study, and importantly if any additional uncertain model parameters go into this wake-merging model?

8. L349: Consider rewording or splitting up the last sentence of this paragraph — "whereas" may not be the appropriate conjunction here. Are there calibrated parameters associated with the pressure feedback?

9. L359: A primary result of the paper is comparing model performance of wind farm blockage effects. In this sense, it seems counter intuitive to justify ignoring dispersive stresses because they are primarily limited to the farm entrance region. How important is this choice and should that be elaborated on in the text? Is the decision to ignore these stresses represented by the model error term?

10. L393: Please check the symbol for the potential temperature.

11. L405: Please clarify the wording in the last two sentences of this paragraph. Is the intention to state that: (1) for the same number of cores, the standard wake model is ten times faster than the APM model, or (2) because the algorithm can be run in parallel, the total simulation time is effectively the same for both models?

12. L410: Consider adding a description for $v_e$ and $v_b$ the first time they are used in the text as the wake model and atmospheric perturbation model parameters, respectively.

13. L414: Intuitively, it would seem the predicted power of the upstream turbines is not sensitive to changes in the wake expansion rate parameters. This sentence seems to suggest the opposite is true. Please double check this sentence. If it is correct, can the authors elaborate on why that is the case?

14. Section 4.1: This section is a nice inclusion of the paper, and provides a clear demonstration of the importance of representing model error in the UQ framework. However it feels out of place in the manuscript and disrupts the discussion between Section 3.3 to 4.2. Would it make sense to put this discussion in Section 2 to motivate representing model error in the Bayesian formulation, or perhaps as an appendix? Please note, this is just a suggestion, and the authors should use their discretion. Either way, nicely done with this example.

15. Figure 8: The posterior distribution suggests negative values of kb may be optimal. Would it make sense to allow negative values of kb, as long as the total wake expansion rate, kw, remains positive? The same considerations may be relevant for Figure 9.

16. L563: "Layouts"

17. Inconsistent usage of "wind farm" versus "wind-farm" throughout the manuscript, including the title.

---

## Author Comment (AC1)

We would like to sincerely thank the reviewers for their careful reading of our manuscript and for providing insightful comments and constructive suggestions. These have been invaluable in improving the clarity and overall quality of our work. Below, we address each point raised in detail.

The Authors

December 19, 2025.

**Reviewer 1**

"Bayesian uncertainty quantification of engineering models for wind-farm atmosphere interaction" provides an in-depth and thoroughly justified approach for an improved UQ method based on authors' previous work. The value is defined clearly, method is well-written and should allow for reproducibility, and credit and reference to other works is sufficiently provided. The paper is well-structured, and figures and tables are used well to provide clarity to points made.

I feel the paper does not require further work, but could be strengthened by attention to the following areas:

**1.1.** Better introduction/description of statistical terms. Non-statistical wind science readers may benefit from a glossary of terms, referral to other sources at points where new terms are introduced, or a brief explanation where some of these terms are provided for the first time.

> We have tried to improve the readability of Section 2 by making notational conventions explicit, reducing the number of introduced symbols, introducing the symbols directly, and writing 'name symbol' instead of the symbol alone.

**1.2.** Similarly, wind science readers with some statistical knowledge may well have only carried out frequentist UQ without being aware of the differential between Bayesian and frequentist statistics, or may not know why it is appropriate here. Giving more reference to this difference in approaches, or clearly outlining the link between Gaussian methods and Bayesian UQ may be useful for readers

> We added a sentence in the introduction that establishes the main difference between Bayesian and frequentist statistics and the advantage of using a Bayesian method in this case: "In Bayesian UQ, the model parameters and sources of uncertainties are characterized by probability distributions, which represent our knowledge about them and which can be updated when more data becomes available."

**1.3.** Subfigure labels are difficult to see, if these could be made clearer (if journal conventions allow)

> We darkened the color of the subfigure labels and increased the font size in Figure 3.

**1.4.** New line equations could be explicitly labelled e.g. a=b /newline = c should be equation 1a, 1b, as they can be considered a new statement (if journal conventions allow)

> We have now explicitly labelled Equations 3, 4, 14, and A9 as 3a, 3b; 4a,4b,4c; etc.

**Reviewer 2**

The manuscript "Bayesian uncertainty quantification of engineering models for wind-farm atmosphere interaction" by Aerts et. al. presents a Bayesian UQ framework for calibrating and comparing wind farm flow models. The Bayesian formulation represents both model error and measurement error and is demonstrated for two engineering models given LES data of a wind farm situated in a conventionally neutral boundary layer. It is shown that, for this dataset, the Bayesian framework quantifies the reduced model error in the higher fidelity model when compared to the lower fidelity model, as well as the uncertainty in each model parameter. Overall the paper is well written, and the examples and figures are effective at demonstrating the utility of the Bayesian UQ framework. High-level comments and line-specific comments are enumerated below.

**2.1.** Please ensure that all symbols and variables are defined in the text throughout the manuscript. I found myself inferring the meaning of several statistical quantities, which may be difficult for the general wind energy audience. A nomenclature section at the beginning of the manuscript would greatly help to enhance the readability of the paper.

> We have tried to improve the readability of Section 2 by making notational conventions explicit, reducing the number of introduced symbols, introducing the symbols directly, and writing 'name symbol' instead of the symbol alone.
> In addition, by moving the analytical example to Section 2, all statistical symbols are now defined and used in Section 2 (except the parameters $\vartheta_e, \vartheta_b$ that are estimated), which may help the readability.

**2.2.** The authors highlight the importance of assessing model uncertainty to provide a robust evaluation of AEP predictions from low-fidelity engineering models, particularly for applications such as wind farm flow control and layout optimization. In wind farm flow control, AEP is often estimated using engineering models (such as the standard wake model considered in this study) across a range of control parameters, with typical AEP

gains over baseline controls of around 1% or even less observed. Given the methodologies and results presented in this manuscript, can the authors provide any insight into whether such small predicted power gains are at all meaningful in light of the large uncertainty and model error observed with the low-fidelity engineering model, such as those shown in Figure 6? Is this error something that can be improved with more data, or are the deficiencies in the model simply too large for low-fidelity engineering models to make meaningful wind farm planning decisions? While rigorously addressing these questions may extend beyond the scope of this manuscript, commenting on them in the conclusion or suggesting them as directions for future work could provide a valuable perspective for the wind energy industry.

> In Figure 6 it is seen that the model error and uncertainty is particularly large for the considered dataset of wind-farm blockage effects in conventionally neutral boundary layers. To obtain an estimate of the model error and uncertainty on the AEP, one needs to average over all atmospheric conditions that occur over a year. We do not expect that the model uncertainty is that large for all conditions, and it is possible that errors cancel out when averaging over a year, so that the AEP uncertainty may be smaller. This is also mentioned in Section 4.3, where we emphasize that the obtained parameters and model error distribution depend on the considered atmospheric conditions.
> We do agree that wind farm flow control and layout optimization studies should compare the obtained power gains to the uncertainty bounds of their model. However, it is difficult to speculate on what these bounds would be for previous studies. Therefore we added the following sentence to the conclusion: "Moreover, the quantified model uncertainty may help evaluate the significance of the obtainable power gains with the controller or optimized layout in the benchmarking conditions."

**2.3.** L44: Define the first instance of "UQ', and please ensure all acronyms are defined the first time they are used.

> We have defined the first instance of 'UQ', and verified that all other acronyms are defined the first time they are used.

**2.4.** L185: What is the justification of using the 0.1 value? Can this be clarified in the text?

> A prior distribution for the model error standard deviations on the turbine power $\sigma_{B,\zeta(i)}$ with an average of 0.1 is deemed weakly informative given that the power of an undisturbed turbine is normalized to be one. Since the posterior distribution of $\sigma_{B,\zeta(i)}$ is expected to be dominated by the likelihood, the exact value should have a small influence.

> We clarified this in the text as: "As the exponential distribution has maximum entropy among all nonnegative continuous distributions with the same average displacement (McElreath, 2018), the standard deviations of the model error terms are assigned exponential priors with averages of 0.1. This is deemed weakly informative given that the power of an undisturbed turbine is normalized to one."

**2.5.** L320: In standard wake models such as an empirically tuned Gaussian model like the one described in Section 3.1.1, it is common to introduce a multiplication factor to scale the inflow wind speed or power coefficient of the turbine to account for the turbine induction field, blockage effects, and other discrepancies with BEM theory. Is such a scaling factor included here? Would that improve the performance of the standard wake model in Section 4, particularly the large modeling error observed in the first row of turbines in Figure 6?

> We wanted to demonstrate that the improved Bayesian UQ works in a context with substantial model error, by comparing a wind farm flow model that accounts for blockage effects with one that does not. For this reason, the wake model does not contain any multiplication factor to scale the inflow speed or power coefficient to account for blockage effects. Since the blockage effect in this dataset is mainly caused by atmospheric gravity waves, we opted for the APM as blockage correction model. It would be interesting to also compare other blockage models, but that would go beyond the scope of the current study. To clarify this point, we added
> "The two wind farm flow models we will consider in this study are a standard wake model (Sect. 3.1.1) and an atmospheric perturbation model (APM) (Sect. 3.1.2). The wake model cannot capture wind farm blockage and is expected to exhibit large model uncertainty for the considered dataset. The second model consists of a wake model with an APM-based blockage correction. Since the blockage effect in this dataset is mainly caused by atmospheric gravity waves, we opt for the APM as blockage correction model, but different models exist (Branlard et al., 2020)."
> We also did not account for the turbine-scale induction effects nor other discrepancies with BEM theory. However, as the turbine power is normalized by the power of an isolated turbine in Figure 6, including such effects would not change the model error in that figure.

**2.6.** L327: This sentence may give the false impression that the wake expansion coefficients are fixed in this study to values of ka = 0.3837, kb = 0.003678. Could this be reworded to clearly state that these are values found in a previous study, and that the wake expansion coefficients will be included as uncertain parameters for the Bayesian

UQ analysis?

> "Here, $k_w$ is the wake expansion rate which is fitted to the local turbulence intensity at turbine as $k_w = k_a I + k_b$ with $k_a = 0.3837, k_b = 0.003678$ (Niayifar and Porté-Agel, 2016)." is replaced by "Here, $k_w$ is the wake expansion rate, which is related to the local turbulence intensity at turbine with two empirical parameters as $k_w = k_a I + k_b$, where a previous fit yielded $k_a = 0.3837, k_b = 0.003678$ (Niayifar and Porté-Agel, 2016)."

**2.7.** L334: Can additional information be added to this sentence to briefly describe the wake merging method used in this study, and importantly if any additional uncertain model parameters go into this wake-merging model?

> We added the requested information: "We will use the one developed by Lanzilao and Meyers (2022), which uses a self-similarity argument to combine the wake deficits through multiplication, without introducing any empirical parameters."

**2.8.** L349: Consider rewording or splitting up the last sentence of this paragraph — "whereas" may not be the appropriate conjunction here. Are there calibrated parameters associated with the pressure feedback?

> We changed this sentence to "Including this pressure feedback allows the APM to simulate the interaction between the ABL flow and gravity waves explicitly, thereby modelling blockage effects without introducing tuning parameters. The farm thrust and turbulent momentum flux, on the other hand, do contain parameters that will be calibrated with the Bayesian framework.".

**2.9.** L359: A primary result of the paper is comparing model performance of wind farm blockage effects. In this sense, it seems counter intuitive to justify ignoring dispersive stresses because they are primarily limited to the farm entrance region. How important is this choice and should that be elaborated on in the text? Is the decision to ignore these stresses represented by the model error term?

> The main objective of the manuscript is to demonstrate the Bayesian UQ framework in a controlled context of large model error, not to quantify the model uncertainty of the currently most accurate version of the APM. The dispersive stresses have a relatively small contribution to the turbine induced forcing, but evaluating them is the most expensive part of an APM evaluation. Hence, the main reason for not including them is the small improvement in accuracy compared to the large increase in

> computational cost. Therefore, we rewrote this line
> "This filtering operation must be applied to the momentum equations as a whole, resulting in dispersive stresses (Devesse et al., 2024a). However, computing the resulting dispersive stresses with the current parametrization is the most expensive part of an APM evaluation (Devesse et al., 2024b). Since these stresses are a minor perturbing force compared to the turbine thrust, we will ignore them to reduce the computational cost. Faster parametrizations for the dispersive stresses in the APM are a topic of ongoing research."
> The model error term is agnostic of the modelling errors made, because the exact source of error is not known in practice.

**2.10.** L393: Please check the symbol for the potential temperature.

> Thank you for noticing this typo. We corrected $\vartheta$ to $\theta$ for the potential temperature.

**2.11.** L405: Please clarify the wording in the last two sentences of this paragraph. Is the intention to state that: (1) for the same number of cores, the standard wake model is ten times faster than the APM model, or (2) because the algorithm can be run in parallel, the total simulation time is effectively the same for both models?

> The intention is to state that the parallel speed-up of the algorithm makes it feasible to perform Bayesian UQ with the APM, but does not reduce the computational cost. For the same number of cores, the wake model (0.83 minutes on one core for 1000 evaluations) is 600 times faster than the APM (8.3 hours on one core for 1000 evaluations). Therefore, sampling the posterior for the APM is 600 times more expensive than for the wake model in terms of compute, for the same number of Monte Carlo samples of their model parameters.
> We clarified the sentence: "Although the SMC algorithm allows performing the many required APM evaluations in parallel to achieve a speed-up in time, the total computational cost remains the same."

**2.12.** L410: Consider adding a description for $v_e$ and $v_b$ the first time they are used in the text as the wake model and atmospheric perturbation model parameters, respectively.

> We clarified this sentence: "An overview of the marginal priors for the empirical model parameters $\vartheta_e$ of the wake model and the APM, as well as the parameters $\vartheta_b = \{\delta_\zeta, \sigma_{B,\zeta}\}_{\zeta=0}^{\zeta_{max}}$ that characterize the model error distribution is given in Table 1." The calibrated empirical parameters for both models are specified in the next paragraph.

**2.13.** L414: Intuitively, it would seem the predicted power of the upstream turbines is not sensitive to changes in the wake expansion rate parameters. This sentence seems to suggest the opposite is true. Please double check this sentence. If it is correct, can the authors elaborate on why that is the case?

> Thank you for noticing this typo. We corrected the sentence: "For the wake model, we know a priori that the sensitivity of the predicted power to the wake expansion rate is only zero for the upstream turbines, which are not waked."

**2.14.** Section 4.1: This section is a nice inclusion of the paper, and provides a clear demonstration of the importance of representing model error in the UQ framework. However, it feels out of place in the manuscript and disrupts the discussion between Section 3.3 to 4.2. Would it make sense to put this discussion in Section 2 to motivate representing model error in the Bayesian formulation, or perhaps as an appendix? Please note, this is just a suggestion, and the authors should use their discretion. Either way, nicely done with this example.

> Thank you for the valuable suggestion. We put the example in between Section 2.2 and 2.3 to improve the continuity of the text from Section 3 to 4.

**2.15.** Figure 8: The posterior distribution suggests negative values of kb may be optimal. Would it make sense to allow negative values of kb, as long as the total wake expansion rate, kw, remains positive? The same considerations may be relevant for Figure 9.

> Imposing that $k_w = k_a I_{min} + k_b > 0$ or $k_b > -k_a I_{min}$ requires specifying a minimal allowable turbulence intensity, which is in principle $I_{min} = 0$. Since the wake expansion should be positive even if the inflow is laminar, we require that $k_b > 0$. To clarify this point, we added "These marginal priors for $k_a$ and $k_b$ ensure that the wake expansion rate is positive for all turbulence intensities." to Section 3.3.
>
> We also corrected Figure 8 to not show postprocessed contourlines for $k_b < 0$, since there are no samples with $k_b < 0$, as indicated by the color shading.

**2.16.** L563: "Layouts"

> We corrected the typo.

**2.17.** Inconsistent usage of "wind farm" versus "wind-farm" throughout the manuscript, including the title.

We changed "wind-farm" to "wind farm" in all compound words.